# Simulating transport and distribution of marine macro-plastic in the Baltic Sea

**Asbjørn Christensen**[1]*, **Jens Murawski**[2], **Jun She**[2], **Michael St. John**[1]

**1** DTU Aqua, Technical University of Denmark, Lyngby, Denmark, **2** Department of Research and Development, Danish Meteorological Institute, Copenhagen, Denmark

* asc@aqua.dtu.dk

## Abstract

We simulated the spatial distribution and dynamics of macro plastic in the Baltic Sea, using a new Lagrangian approach called the dynamical renormalization resampling scheme (DRRS). This approach extends the super-individual simulation technique, so the weight-per-individual is dynamic rather than fixed. The simulations were based on a mapping of the macro plastic sources along the Baltic coast line, and a five year time series of realistic wind, wave and current data to resolve time-variability in the transport and spatial distribution of macro plastics in the Baltic Sea. The model setup has been validated against beach litter observations and was able to reproduce some major spatial trends in macroplastic distributions. We also simulated plastic dispersal using Green's functions (pollution plumes) for individual sources. e.g. rivers, and found a significant variation in the spatial range of Green's functions corresponding to different pollution sources. We determined a significant temporal variability (up to 7 times the average) in the plastic concentration locally, which needs to be taken into account when assessing the ecological impact of marine litter. Accumulation patterns and litter wave formation were observed to be driven by an interplay between positive buoyancy, coastal boundaries and varying directions of physical forcing. Finally we determined the range of wind drag coefficients for floating plastic, where the dynamics is mostly directly wind driven, as opposed to indirectly by surface currents and waves. This study suggests that patterns of litter sorting by transport processes should be observable in many coastal and off-shore environments.

## Introduction

Floating plastic debris in the marine environment is an increasing problem [1] and litter cleanup and emission management are demanded by stakeholders, the general public and governing bodies, despite potentially high costs for modest gains. To make qualified decisions to prioritize the use of limited economic resources to this end, it is critical to make choices based on scientifically grounded decision support tools [2, 3]. Even though marine litter is clearly a visible problem along the Baltic Sea coastline, the relation between sources, transport processes and deposition has only recently been investigated for microplastics [4] and studies are lacking

**Data Availability Statement:** All code developed in this work is available at https://github.com/IBMlib/IBMlib, as detailed in supplementary information. This include input scripts files to run baseline simulations and data set for plastic pollution

sources Third party input underlying the results presented in the study are available from respective providers: wind and wave fields: ECMWF (www. ecmwf.int/); hydrographic data: DMI (www.dmi. dk).

**Funding:** This work was supported by the European Commission (https://ec.europa.eu) under the Horizon2020 programme as project CLAIM (Cleaning Litter by developing and Applying Innovative Methods in european seas), grant agreement No. 774586. The contribution of all authors (AC,JM,JS,MstJ) were funded by this grant. No commercial companies supported this work by funding. The funder had no role in study design, data collection and analysis, decision to publish, or preparation of the manuscript.

**Competing interests:** The authors have declared that no competing interests exist.

for larger plastic fractions despite evidence [1] that it deteriorates habitat quality and can cause direct harm to wild life in the Baltic Sea ecosystem.

To this end, a better understanding of how plastic debris is transported from coastal and marine sources to ecologically sensitive habitats and recreational areas is crucial to design and implement mitigation actions [5]. The study of marine plastic debris transport has been spearheaded by several studies performed at global scale. These early works studied the formation and long-term dynamics of garbage patches in subtropic Ekman convergence zones, identified new potential aggregation zones [6] and the scale for aggregation dynamics [7]. Furthermore, these studies emphasized the importance of using properly weighted source distributions to obtain realistic transport dynamics and equilibrium distributions [8].

All these studies were of the Lagrangian type [9] where a representative set of litter objects was tracked by integrating equations of motion forward in time by employing climatological (time-averaged) drift-velocity maps, constructed from several years of buoy track data or temporally cycled circulation model time series, in order to conduct multi-decadal simulations of litter transport. Simulation of marine litter transport poses a new challenge to physical atmospheric/ocean circulation models. Representation of small scale processes, ocean-air interaction and the ocean-air interface are of particular importance for simulating plastic transport [5]. Historically skill validation for circulation models has focused on tangible ocean bulk outputs like vertical profiles and single point in situ time series and ocean surface properties aligned with remote sensing data. Here exceptional events, like the Great Japan Tsunami in 2011, constitute a unique opportunity to assess the uncertainty of parameters in models of litter transport [10].

In addition to global scale litter transport studies, regional studies are emerging, taking advantage of recent progress in and availability of high quality operational hydrography at local scale. In the North Sea for example, [11] found a seasonal signal in the number of tracer particles that reached the coastal areas, but this study could not identify accumulation regions at open sea. In the Mediterranean, [12] found a general tendency of floating matter to collect in the southern portion of the basin, and in particular a long term accumulation in the southern and southeastern Levantine basin. In the Sea of Japan [13] examined transport of a particular plastic item (lighters) and found a residence time of less than 3 years unless beached in this regional sea, which is relatively open and connected to the East China Sea, Sea of Okhotsk and the Pacific Ocean at several points. The focus on a particular, well-defined narrow litter fraction removes some parameter uncertainty and makes the comparison with observational data more stringent, even though it prunes the available data for comparison.

For regional scale studies transport pathways and time scales are shorter, since regional pollution sources are likely most important and of interest for developing mitigation options. Further regional cases are often open source-sink systems, not closed source-sink systems as in global scale studies, hence these need to be studied with appropriate boundary conditions. Furthermore, regional scale studies pose further challenges due to the necessity for proper representation of mesoscale features and their resolution in circulation models. Additionally, the coastal zones are more important; even though global circulation models have been steadily evolving in terms of complexity, horizontal resolution, and process parameterization, coastal and shelf phenomena are still poorly replicated or even misrepresented as the grid mesh is often too coarse. This is especially true for complex-geometry regions such as sea straits, archipelagos, or semi-enclosed seas where topographic details are not well resolved [14]. Furthermore litter-beach interaction becomes relatively more important on the regional scale with a larger coast/ocean ratio and weight of coastline complexity. Litter-beach interaction is a relatively new research area with large uncertainty in process knowledge, despite recent advances

[15–18]. Much is to be learned from related research areas involving beach interaction processes, e.g. oil spill modelling [19] and sediment dynamics [20]. To alleviate the potential shortcomings of regional scale operational hydrography provided by circulation models, which are typically augmented by data-assimilation to enhance model skill level, regional scale transport studies may certainly benefit from envisioned future integrated marine debris observing system (IMDOS) [21]. IMDOS has the potential to provide data fusion of multiple sources and may correct for unresolved physics and biases in current circulation models. However, the time horizon for access to the envisioned IMDOS for the wider scientific community is unclear.

Lagrangian simulations have two main flavors: fixed-identity, where a computational particle represents one real object or the super-individual approach [22], where a computational particle represents a specified number of objects lumped together. Technically, the numerical resolution of Lagrangian simulations is best, if the largest feasible number of computational particles are employed in the simulation, since too few computational particles can lead to loss of variation, irregular dynamics, and large sensitivity to the value of random generator seeds [22], since the resolution accuracy increases quite slowly only as $1/\sqrt{M}$, where $M$ is the number of computational particles in the simulations.

The objective of the present work is to demonstrate a variant of the Lagrangian super-individual approach with variable normalization and fixed number of computational of particles that is efficient for modelling an open source-sink system and apply this approach to gain new insights into the dynamics and distribution of floating marine plastics in the Baltic Sea.

## Materials and methods

### Study area: The Baltic Sea

The Baltic Sea is an Eastern marginal sea of the of the Atlantic Ocean bordered by nine nations. The sea stretches from 53˚N to 66˚N latitude and from 10˚E to 30˚E longitude. The Baltic Sea is a strongly stratified semi-enclosed basin and one of the worlds largest brackish inland seas by area, with a net surface outflow driven by large freshwater supply from rivers and occasional deep layer high salinity inflows through the straits of Denmark. We have indicated the average large-scale circulation in the Baltic Sea in Fig 1. The Baltic Sea is dominated by westerly winds averaging to 7.5 m/s and the ratio of westerly winds to easterly winds is about 18:11, so there is significant directional variability [23]. Corresponding spatial and temporal variability in wave height and direction is also observed [24]. An in-depth review of the variability and complexity of the Baltic Sea physical oceanography may be found in [25], along with the challenges of modelling these.

The southwestern Baltic Sea is well oxygenated and have a high biodiversity, whereas the remainder of the Sea is more brackish and poor in oxygen and displays a decrease in species richness from the Danish belts to the Gulf of Bothnia. Approximately a quarter of Baltic seafloor is a variable dead zone due to hypoxy caused by excess nutrient loading in combination with stratification and therefore the seafloor ecology differs from that of the neighboring Atlantic. The Baltic Sea ecosystem is a mixture of marine and freshwater species. To protect the valuable marine and coastal habitats in the Baltic Sea, a network of marine protected areas (HELCOM MPAs and Natura 2000 areas) have been established. Yet, these areas are subject to many antropogenic stressors, e.g. shipping, eutrophication and pollution. The exposure of these protected areas to marine litter pollution has recently been addressed in other works [3, 27].

a)

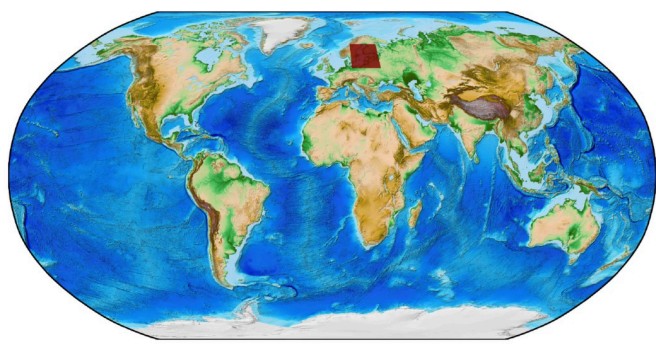

b)

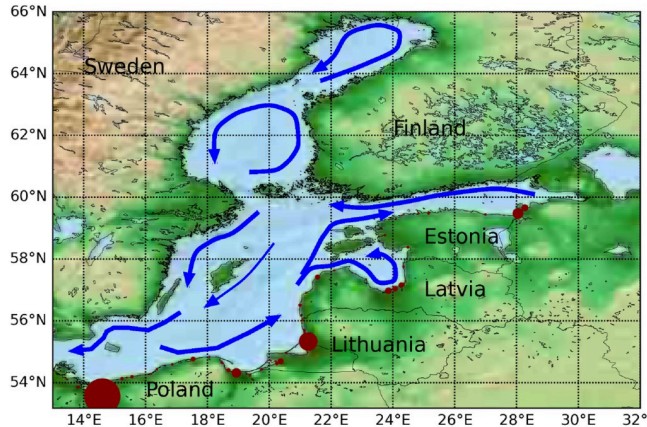

**Fig 1. Study area and river sources.** (a) Model domain indicated by read polygon. (b) Model domain with river sources of macroplastic inflow to the Baltic indicated with circle diameter scaled according to yearly influx. The largest source, river Oder, in lower left corner correspond to 67 tons macro plastic per year. Blue arrows show a schematic view of time-averaged large-scale surface circulation (reproduced from [26]).

## Baltic Sea physical model

The physical data used in the transport simulations is produced by the Baltic-North Sea ocean-ice model HBM (HIROMB-BOOS Model) in the operational setup by DMI (Danish Meteorological Institute). The model has been jointly developed by the HBM consortium and used as an operational model in Denmark, Estonia, Finland and Germany. HBM is a three-dimensional, free-surface, baroclinic ocean circulation and sea ice model that solves the primitive equations for horizontal momentum and mass, and budget equations for salinity and heat on a spherical grid. Vertical transport assumes hydrostatic balance and incompressibility of sea water. Horizontal advection is modelled using a flux corrected transport scheme with the Boussinesq approximation applied. Higher order contributions to the dynamics are parameterized following [28] in the horizontal direction and a k-$\omega$ turbulence closure scheme, which has been extended for buoyancy-affected geophysical flows in the vertical direction [29, 30].

The model allows for fully two-way nesting of grids with different vertical and horizontal resolution, as well as time resolution, to resolve narrow straits and channels. The numerical model implementation uses a staggered Arakawa C-grid and z-level coordinates and free-slip conditions along the coastlines. With two-way dynamical nesting, HBM enables high resolution in regional seas and very high resolution in narrow straits and channels. With its support for both distributed and shared memory parallelization, HBM has matured as an efficient and portable, high quality ocean model code. The HBM setup for the present hydrographic dataset has a horizontal grid spacing of 10 km in the North Sea and in the Baltic Sea, and 2 km in the inner Danish waters. In the vertical the model has up to 50 levels in the North Sea and the Baltic Sea, and 52 levels in the inner Danish waters with a surface layer thickness of 2 m. HBM is forced by DMI-HIRLAM with 10 m wind fields, sea level pressure, 2 m temperature and humidity and cloud cover. At the open model boundaries between Scotland and Norway and in the English Channel, tides composed of the 8 major constituents and pre-calculated surges from a barotropic model of North Atlantic [31] are applied. Other variables are set to monthly climatological values. Freshwater runoff from the 79 major rivers in the region are obtained from a mixture of observations, climatology (North Sea rivers) and hydrological models (Baltic Sea). At the surface the model is forced with atmospheric data from the numerical weather prediction model HIRLAM [32]. HBM performance has been validated on several occasions, e.g. [29, 33–37]. The HBM model is validated on an annual basis as DMI's operational storm surge model. It has been extensively validated as CMEMS Baltic marine Forecasting model until 2020 [38] and as operational model for coastal applications [39] in these basins.

## Baltic Sea macro litter sources

Riverine and other sources of macro litter along the Baltic coastline have been mapped in the CLAIM project [40], as reported in [41], and this input is used to simulate the dynamics of litter distribution and transport. The river sources are shown in Fig 1, with circle diameter scaling with input flux. In total, the litter influx is represented by 21464 point sources along the Baltic coast line. The spatially uneven distribution of river loads make it very important to account for these explicitly; the six largest river sources together account for 93% of the river load of macro plastic into the Baltic Sea. The input data is split into two parts: riverine and coastline sources. Riverine sources are provided in units tons/year and is aggregated into 402 point sources. The coastline input from [41] is in unit items/coastlength/year, provided as 21062 uniformly distributed sampling points along the Baltic coastline. Each such loading point is projected onto the beach line corresponding to the hydrodynamic data set to have a consistent representation. To make coastline and riverine input commensurable, a conversion scale average weight/item (w) is needed. Significant uncertainty is associated with estimating this scale. We determine that a threshold value w = 4 g/item makes total riverine and coastline input equal in this data set. When w is larger than 4 g/item, coastline input exceeds riverine input. We apply the estimate w = 10 g/item applying to Mediterranean litter input [42] in our simulations, but some recordings could suggest a higher value for w. Furthermore in this study the Russian coastal input has not been accessible for the source mapping and as result is not addressed in this study.

## Baltic Sea macro litter transport model

The continuous litter distribution necessary to generate maps of macrolitter dynamics were generated by Lagrangian simulations and averaging particle distributions over a 10 km scale (corresponding to the scale of the hydrodynamic model applied). The modular Lagrangian framework IBMlib, which has been used in numerous studies of physical-biological

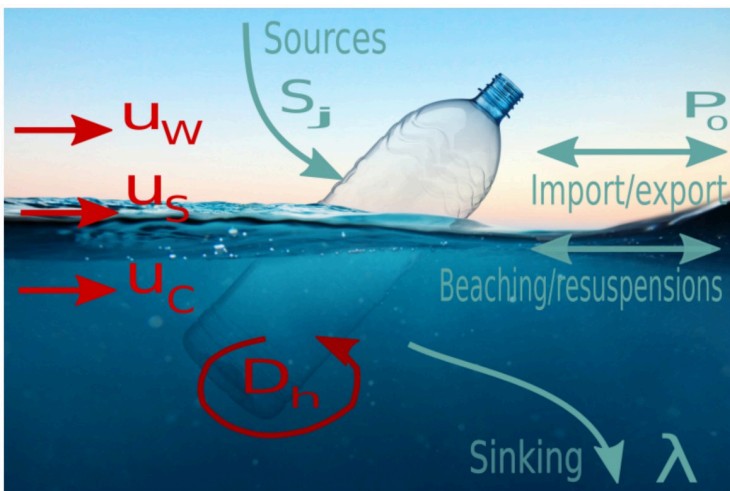

**Fig 2. Sketch of key processes included in the physical transport model.** $u_w$, $u_s$, $u_c$ are wind, Stokes and surface current transport, $D_h$ are horizontal diffusivity from small-scale eddies, $S_j$ are coastline sources, $P_0$ are plastic concentration at the open boundaries, and $\lambda$ the local sinking/removal rate of plastic.

interactions [43], many of which have been conducted in the North Sea / Baltic ecoregion, were used for the offline simulations, employing a stored HBM hindcast database with 10 km horizontal resolution, 1h temporal resolution and 50 vertical layers in a z-grid configuration, as described above. All major physical processes recognized to be important for horizontal and vertical transport of visible plastics are included in the setup; a sketch of included processes are provided in Fig 2 These include advection by ocean currents $u_c$, obtained from the HBM model, and Stokes drift $u_S$ from ECMWF ERA5 atmospheric reanalysis [44] with 1h time resolution. Wind drag on low-density plastic objects generates a drift velocity $u_w$ which can be argued [13] to be linear with the 10 meter air velocity vector $u_{10}$ also obtained from the ECMWF ERA5 atmospheric reanalysis with 1h time resolution

$$u_w \sim k_0 \sqrt{A_a/A_w} u_{10} \sim k_0 \sqrt{\frac{\rho_w}{\rho_p} - 1}\ u_{10} = k\ u_{10} \tag{1}$$

$A_a$ is the area perpendicular to the wind direction of the plastic object above the sea surface, and $A_w$ the area below the sea surface. $\rho_w$, $\rho_p$ are the densities of water and plastic, respectively, and $k_0$ is a heuristic shape factor of order 0.03 [13], consistent with [10], expressing the ratio between above/below surface drag coefficients and air/water density as well as effective wind vertical profile near the sea surface. In addition to this comes surface layer wind drag, which in hydrodynamic models is averaged over upper grid cell; however floating plastics experience only the skin layer, which is estimated to be 4% of $u_{10}$ above the layer vertical average, based on a log-scaling estimate, which is added to the windage term. We neglect the 45° inclination of this term of skin drift to avoid $k$ being a matrix, as a consistent treatment would require removing the average Ekman spiral from the upper hydrodynamic layer consistently with the HBM representation. A matrix representation of $k$ needs separate investigation and validation. Further, for simplicity we assume the area above ($A_a$) and below ($A_w$) the sea surface being equal, since the uncertainty is absorbed in $k$. We later assess the influence of this by considering a range of $k$. Consequently in the present baseline runs, we apply $k = 0.07$, representing pure windage and correction for finite upper layer thickness in the circulation model. This is in good agreement with [13] who estimated the relevant windage range to be $0 \le k < 0.3$, and

[11] who found a good match with data when applying $k \sim 0.05$. To account for horizontal sub-grid scale eddy diffusivity $D_h$, we apply the standard shear-driven Smagorinsky scheme [28] dynamically *a posteori* on hydrodynamic current fields $u_c = (u, v)$:

$$D_h = \frac{C_S^2 \Delta_x \Delta_y}{Sc} \sqrt{\left(\frac{\partial u}{\partial x} - \frac{\partial v}{\partial y}\right)^2 + \left(\frac{\partial u}{\partial y} + \frac{\partial v}{\partial x}\right)^2} \tag{2}$$

where conventional values [45] for the Smagorinsky constant $C_s = 0.2$ and Schmidt number $Sc = 1$ were applied, and $\Delta_x \sim \Delta_y \sim 10$ km are the horizontal grid resolutions.

Altogether the particle positions $x$ are updated by the finite stochastic differential

$$dx = (u_c + u_s + u_w)dt + \nabla D_h dt + \eta \sqrt{\frac{2}{V} D_h (x + \frac{1}{2} \nabla D_h dt)dt} \tag{3}$$

where $dt$ is the time step of 1800 seconds and $\eta$ is a random variable uniformly distributed on [-1,1] corresponding to variance $V = 1/3$. The first term is the deterministic advective term. The third term emulates a random walk corresponding to $D_h$ [46] and the second term prevents artificial particle aggregations [46] when $D_h$ is spatially inhomogeneous. The simple Euler-forward scheme is used to solve Eq 3, as popular Runge-Kutta algorithms lose their time step advantage for stochastic equations of motion. Time step is chosen so that step size Eq 3 is well below the finest scale of forcing data grids and that the advective trajectories ($D_h = 0$) has negligible integration error from $dt$. Physical aggregation may occur in the basin interior, because neither advection field $u_s$ or $u_w$ are divergence free; further, in combination with positive buoyancy, $u_c$ is not horizontally divergence free, because downward currents may form aggregation zones, as observed in relation to the infamous persistent garbage patches [47] Further aggregation at coastal boundaries may occur, because advective components $u_s$, $u_w$ does not vanish at coastal boundaries.

Natural processes are ultimately removing plastic objects from the water column. It is believed that the major processes are fragmentation, biofouling and eventually sinking (or shore burial) [48]. Even though progresses have been made [49] in resolving spatio-temporal variation in removal rate $\lambda$, uncertainty is currently too high [5] and baseline simulations were conducted using a constant value of $\lambda = 0.003$ day$^{-1}$. Retention and reactivation(resuspension) are opposite processes that eventually reach a dynamical equilibrium, so that rates are equal and opposite when averaged over medium to long time scales. Both processes are currently not well parameterized and subject to ongoing research. In order to avoid additional submodels without strong observational support, we assume that the dynamical equilibrium between retention and reactivation applies at short time scales for Baltic macro litter simulations. Technically this implies that reflective boundary conditions apply along coastlines to incoming litter. Since the hydrographic data set is on a regular longitude-latitude grid, the corresponding consistent coastline is constituted of zonal and meridional segments of approximately same length. In this geometry, reflective boundary conditions for Eq 3 implies solving a multiple-reflection problem of an arbitrary step Eq 3 crossing the coastline, and this can be solved exactly and is implemented in IBMlib. This is important to avoid potential fine-scale coastal particle aggregation by *ad hoc* schemes for particle-coastline interaction. The key parameters for the Lagrangian setup are summarized in Table 1.

## The DRRS scheme for quasi-equilibrium distributions

Lagrangian simulations have the ability to resolve the influence of a huge number of sources at the same time, because particles in the simulation can carry a source label without overhead, as

**Table 1. Key parameters for the baseline Lagrangian simulations.** References are provided in text along with the description.

| Parameter | symbol | value |
|---|---|---|
| Simulation period | | 2008–2012 |
| Equilibration period | | 2008 |
| Time step of Eq 3 | dt | 1800 s |
| Removal rate | $\lambda$ | 0.003 year$^{-1}$ |
| Windage coefficient (Eq 1) | k | 0.07 |
| DRRS smoothing time | T | 30 days |
| Particle ensemble size | M | $10^5$ |
| Boundary concentration of plastic | $P_0$ | 0 |

compared to Eulerian simulations which need a costly 2D/3D field variable for each source. On the other hand, since the number of floating plastic objects in e.g. the Baltic ecoregion by any estimate far exceeds $10^8$, each particle (in the literature also called individual, agent or ant) in a Lagrangian simulation must represent several objects, thereby becoming so-called super-individuals, since a laptop simulation currently only can handle $10^5$-$10^7$ particles, depending on included processes and simulation length. A Lagrangian simulation generally tries to include as many particles $M$ as possible, if all real objects cannot be included in the simulation, since the simulation estimate uncertainty decreases with $M$. Some considerations must be made to utilize the computational capacity of $M$ particles optimally. In the classical super-individual approach [22] in Lagrangian simulations each particle in a simulation represents a fixed number of real objects. A naive setup will release particles (corresponding to a fixed number of real objects) to the marine ecosystem at a rate chosen so that over the simulation period the total number of released particles reaches $M$. This is inefficient, because before the system reaches a quasi-equilibrium, a large fraction of particles (plastic objects) must be subject to sinking/loss processes, and therefore inactive in resolving the pelagic spatial distribution which is of interest; this leads to resampling type simulations [50], where sunken/lost plastic objects are reinjected into the particles ensemble. Still this is suboptimal, because it requires a precise (but conservative) estimate of the total number of free particles in water masses, i.e. a fixed conversion ratio between particles and real plastic pieces. This leads us to propose a novel variant resampling scheme for this type of problem, the dynamical renormalization resampling scheme (DRRS). Following this approach, the conversion ratio between particles and real plastic pieces is dynamic and computed on the fly, and the number of free particles is fixed to $M$, to maximize the statistical resolution of the plastic distribution. The computational principle is outlined below in Fig 3. The main advantage is that the conversion between particles and real floating objects is automatized.

The scheme basically relies on the fact that the plastic distribution dynamics $P(x, t)$ is linear with its sources. The key idea in Fig 3 is that all rates and densities from the left (real system) to the right side (DRRS ensemble) are related by a common rescaling factor, the number of litter items each Lagrangian particle represents, which is not fixed in our approach. All particles leaving the system (by sinking or advection across an open boundary) are immediately reinjected into the system at a source with a probability proportional to the real influx from that source, so that the total number of active particles $M$ in the simulation is constant at all times. Beached particles are reinjected locally by the reflective boundary condition, corresponding to the assumed local beaching/resuspension equilibrium applying at medium time scales. This means that the unknown spatio-temporal plastic distribution $P(x, t)$ can be assessed by identifying the ratio of the real influxes to the total resampling rate (determined by internal system

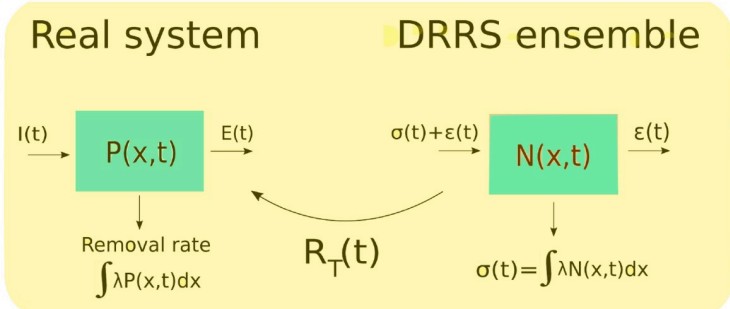

**Fig 3. The DRRS algorithm (dynamical renormalization resampling scheme) for determining absolute (real world) distribution from a fixed-size ensemble of particles.** I refers to plastic influx and E export from the simulated real system. The key idea is that all rates and numbers from left (real system) to right side (DRRS ensemble) are related by a common rescaling factor. The rescaling factor $R_T$ is determined by comparing absolute known source influxes I to resampling rate $\phi = \sigma + \epsilon$ in the simulated system on the right side. T refers to the time scale over which uninteresting stochastic fluctuations are smoothed.

dynamics and transport processes) $R_T(t)$, multiplied by DRRS ensemble density $N(x, t)$, which integrates to the fixed ensemble size $M$ at all times.

$$P(x, t) = R_T(t)N(x, t) \tag{4}$$

Here $N(x, t)$ is constructed by convoluting particle positions $\{x_i(t), i = 1 \ldots M\}$ over space, e.g. simply counting particles per cell on a suitable output grid at time $t$. The rescaling factor $R_T(t)$ is resolved by comparing absolute source influxes into the system to the resampling rate, both of which are accessible in the simulation. Here we need to consider the open boundaries of the system, which are divided in two parts: one, where the instantaneous net advection $u = u_c + u_s + u_w$ is oriented toward the interior of the system $W_+(t)$ and one where the instantaneous net current is oriented toward the exterior of the system $W_-(t)$. The input to the real system is

$$I(t) = \sum_j S_j(t) + \int_{W_+(t)} P_0 \; u \cdot \hat{n} \; dA \quad \geq \quad 0 \tag{5}$$

where $P_0$ is the external plastic concentration at the system boundary and $\hat{n}$ is the boundary normal vector (oriented toward the inside), and $dA$ an integration area element of the system boundary. The first term is input from point sources, where $S_j(t)$ are prescribed influxes (e.g. riverine sources, Fig 1). The second term describes plastic advected into the system at boundary concentration $P_0$. In our case this term is negligible due to the net surface outflow from the Baltic Sea. The export from the real system (Fig 3, left side) is

$$E(t) = -\int_{W_-(t)} P(x, t) \; u \cdot \hat{n} \; dA \quad \geq \quad 0 \tag{6}$$

which describes the plastic advected out of the system across open boundaries. In the quasi-equilibrium state of the system which we are interested in,

$$I(t) \sim E(t) + \int \lambda P(x, t)dx \tag{7}$$

where the last term is the net sinking rate. The right side of Eq 7 corresponds to particles leaving the system (which are reinjected in the Lagrangian system). We denote the rate of particles transported out of the system across open boundaries $\epsilon(t)$ and the rate of particles sinking $\sigma(t)$, and these satisfy $\epsilon(t) \sim E(t)$ and $\sigma(t) \sim \int \lambda P(x, t)dx$ and by Eq 7 $I(t) \sim \epsilon(t) + \sigma(t) = \phi(t)$. If

particles crossing open system boundaries and sinking were smooth processes, then simply $R_T(t) = I(t)/\phi(t)$. However, on the time scale $dt$ used for solving the equation of motion Eq 3, $\phi(t)$ is governed by stochastic fluctuations, and further only the long term average of $I(t)$ is available. Therefore we apply a smoothing operator $\hat{q}_T$ to create a running average

$$\hat{q}_T(t)y = \frac{1}{T}\int_0^\infty e^{-\frac{s}{T}}y(t-s)ds \tag{8}$$

where $y(t)$ is test function used to display how the operator $\hat{q}_T$ works. The operator averages the test function $y(t)$ over a period $T$ truncated smoothly into the past. We therefore apply the running average to solve Eq 4

$$R_T(t) = \frac{\hat{q}_T(t)I}{\hat{q}_T(t)\phi} \tag{9}$$

where the smoothing time scale $T$ is suitably chosen to reduce stochastic fluctuation noise in the renormalization factor $R_T(t)$.

In this way, given absolute values for point source influxes $\{S_j(t)\}$ (which are active during the entire simulation period), the absolute concentration of marine plastic can be assessed by Lagrangian simulations. The computational efficiency gain of the DRRS scheme over the conventional Lagrangian super-individual approach is demonstrated in S3 Benchmarking the DRRS algorithm for a simple reference system, where the analytical solution can be developed. The meaningfulness of the renormalization Eq 4 relies on the establishment of quasi-equilibrium distribution. A quasi-equilibrium state of the system is guarantied, because sources have a prescribed rate, whereas losses scale with the plastic density $P(x, t)$. The beaching/resuspension processes of visible plastic occurring along the coastline are assumed to be in local dynamical equilibrium at the time scales addressed. The scheme gives the correct limit, when sources are constant and a smoothed dynamics when inputs are time varying. In practice we find the renormalization operator in Eq 9 to be quite stable when $T$ is set appropriately. When above a certain threshold it is our experience that $T$ is not a sensitive parameter; we find $T \sim 30$ days is a reasonable choice for the Baltic Sea. In our case, the value for the boundary condition $P_0 = 0$ is not critical, because the Baltic Sea has a net surface outflow.

## Green's functions and plastic distribution

In relation to cleaning efforts it is of paramount importance to identify the sources that contribute to plastic pollution locally. The density of floating plastics in the marine environment is still low enough (most places) such that transport processes can be considered linear; that is, plastic items from different sources are transported independently of each other from their respective sources. Therefore the average concentration (per area) of plastic can strictly and generally be expressed as

$$P(x, t) = \sum_j \int_{-\infty}^t S_j(t')G(x_j, x; t', t)dt' \tag{10}$$

where $G(x', x; t', t)$ is the time-dependent Green's function or transport rate of a plastic piece released at $(x', t')$ to the position-time $(x, t)$, and $S_j(t')$ is the plastic influx from the source $j$ at time $t'$, where we here will just consider the average rate from point to point. Since observations currently only support stationary sources, we initially focus on time-averaged

distributions $P(x) = < P(x, t) >$ and we get

$$P(x) = \sum_j S_j G_j(x) \tag{11}$$

where $G_j(x) = < G(x_j, x; t', t) >_{t',t}$ can be considered to be the average plastic plume from source $j$ at medium time scales (months to years), and the sum is over all relevant sources. Here we advocate to estimate $G_j(x)$ from current, wave and wind-driven transport processes, as direct monitoring will not resolve $G_j(x)$ sufficiently. We address time-variability issues in the discussion. It is essential that calculation of $G_j(x)$ includes export and sink processes, resulting in a medium time scale quasi equilibrium distribution of the plastic. The Green's function assesses depth strata of interest, typically surface for macro litter or the photic zone for micro-litter. If needed, depth in the water column $z$ can be included as well along with $x$. Of special interest is the demersal Green's function $G_j^D(x, t)$, representing the sunken litter (assuming sinking is the dominating terminal process for marine plastic). Here the level is not in equilibrium, but increases over time unless resuspension is considered. For management purposes, it is better to consider the sinking flux $F_j(x) = \lambda G_j(x)$, which establishes equilibrium at same time scale as a horizontal Green's function $G_j$. If sinking processes are just characterized by a (possibly seasonal) time scale, $\lambda$ will be spatial and source independent, giving the same result as the analysis with pelagic Green's function. In the examples in subsequent sections, we use Lagrangian simulations with current, wave and wind-driven transport to assess $G_j(x)$ at a regional scale for the Baltic Sea.

## Baseline simulations

The baseline simulations presented below consists of two parts: (i) average-emission simulations where all Baltic Sea macro litter sources were active and (ii) Greens function simulations where the pollution plume from individual macro litter sources was assessed (i.e. all other litter sources was set to zero, except the one assessed). Both parts were conducted with the setup described in Table 1, so that the simulation period was 5 years (2008–2012 inclusive), with the year 2008 considered to be the transient period for the plastic distribution to settle to the dynamic equilibrium; by this we mean the long-term average distribution plus a fluctuating part reflecting the variability in physical forcing. The dynamic equilibrium distribution is typically settled faster than one year with the DRRS algorithm. Only Greens functions corresponding to a few selected interesting macro litter sources out of 21464 point sources included in (i) are presented. When not stated otherwise, spatial plots below correspond to the time average for the 4 year period 2009–2012, to represent the climatic variability in physical forcing. With respect to Greens function simulations (ii) for a particular litter source using the DRRS scheme, lost or sunken particles in Fig 3 were only reinjected at that particular source.

## Model validation

Validation data with sufficient spatial resolution and temporal span is difficult to obtain but an indication of model skill may be obtained by comparing beach litter observation data with time averages of plastic density $< P(x, t) >_t$ from Eq 4 in coastal hydrodynamic cells. For the Baltic Sea beach litter observation data is provided by the monitoring program [51] coordinated by HELCOM, which is an intergovernmental organization established to protect the marine environment of the Baltic Sea. The monitoring program covers physical, chemical and biological variables of the Baltic marine environment. Beach litter includes consumer plastics (e.g. cigarette butts, food wrappers, beverage bottles, straws, cups and plates, bottle caps, and single-use bags) and lost fishing gear (e.g. rope fragments, polystyrene floats). Here it is

**Fig 4. Log scale comparison of model data and beach litter observations in unit items/km.** Circle location correspond to observation, and color scale litter abundance. (a) Beach litter observations. (b) Model coastal litter prediction, corresponding to a coastal beaching affinity of $a = 6.4 \, 10^5$ km·items/ton.

assumed that beaching/resuspension is in dynamical quasi-equilibrium and consequently that a constant coastal litter affinity $a$ applies: $B(x) \sim a < P(x, t) >_t$, where $B(x)$ is the beach litter concentration (i.e. per length). In Fig 4 we show on log scale the average beach litter density for 2012–2016 density versus $<P(x, t)>_t$ for 2009–2012, corresponding to a coastal litter affinity of $6.4 \, 10^5$ km·items/ton. The data correlation gives a satisfactory $r = 0.48$ and $p = 2.5 \, 10^{-7}$, given the assumption of constant coastal litter affinity, and that some interannual variability is present in beach litter observations [51], and that source set lacks Russian data. Furthermore perpendicular beach width is not recorded in relation to observations. The model is able to reproduce major spatial trends in accumulation increasing on eastward coasts, potentially due to prevailing Western winds, and higher abundance in Bothnian Sea coasts than Southern coasts. Notice that abundance observational data spans more than five orders of magnitude. The model value span in abundance is slightly higher than observed beach litter, but in overall satisfactory agreement.

## Results

### Macroplastic dynamics in the Baltic Sea

Fig 5 shows two typical daily snapshots of the simulated macro plastic distribution in the Baltic Sea, with one month in between, taken from a long term simulation 2008–2012. The snapshot distributions are remarkably inhomogeneous. In Fig 5(a) from Jan 1[st], 2012 litter is concentrated more on Eastern coast lines, with clear patches, whereas in Fig 5(b) from Feb 1[st], 2012, plastic is more abundant along Western coastlines, but less patchy in occurrence. Also away from coastlines, litter abundance fluctuates, with occasional cleaner areas. The coastal aggregation is caused by advective components in Eq 3 not vanishing at the coastal boundaries. When wind/waves shift direction, these aggregations are then transported offshore in apparent garbage waves, which resemble Langmuir circulations even though they have a different scale, typical of the distribution of floating particles due to wind generated circulations in the surface layers. These waves have a long persistence before eventually being dispersed by sub scale eddies or relaxation of wind stress. This boundary molding effect of plastic waves is expected in other closed/semi-closed seas with variable wind conditions, like the Mediterranean and the Black Sea. This means that the local macroplastic concentration fluctuates by many orders of magnitude, which is important to include in ecological impact analyses. To give an overview of the spatial pollution distribution, we display in Fig 6 selected long term and seasonal averages for a simulation 2008–2012, starting from randomly distributed litter, with 2008 being the spin-up period. Riverine and other sources of macro litter are active at constant level during the entire simulation period.

The long term average Fig 6(a) covers 2009–2012, and we see a tendency for coastal accumulation, which is more pronounced on Eastern coast lines, higher average concentrations in

a)

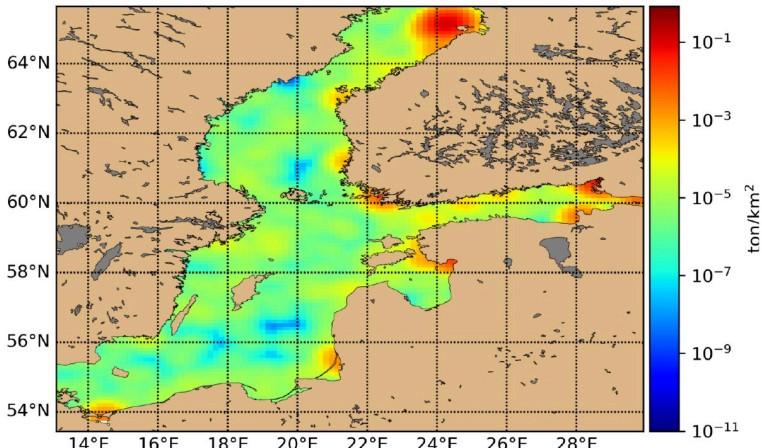

b)

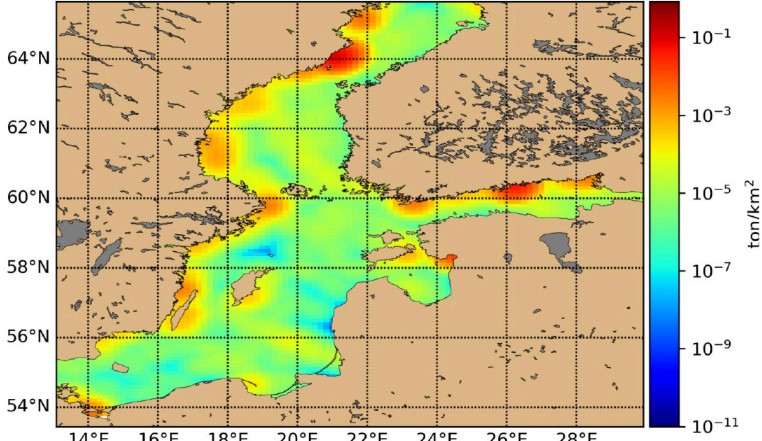

**Fig 5. Typical time snapshots of the simulated macro plastic distributions (tons/km²) in the Baltic Sea corresponding to a) Jan 1ˢᵗ, 2012 and b) Feb 1ˢᵗ, 2012.**

the Bothnian Sea than Southern parts of the Baltic Sea. Since the Baltic Sea surface currents at the entrance to the Baltic Sea are generally an outflow, most litter not removed is exported to the Danish transition zone, with a concentration gradient toward the entrance. Currently, no data is available for import via the Danish transition zone, which may decrease the gradient toward the Baltic entrance at the map, but the effect is assumed small, as the Baltic Sea is an outflow zone transporting freshwater discharge from Central and Eastern Europe.

Fig 7 shows the time statistics for the period 2009–2012 of two West-East transects at 56˚N and 62˚N, respectively. The transects show the overall trend that pollution increases toward the North. Further it shows that in a coastal zone of width 10–50 km, the average concentration is elevated by a factor 3–6 compared to offshore at similar latitude. Finally, Fig 7 shows

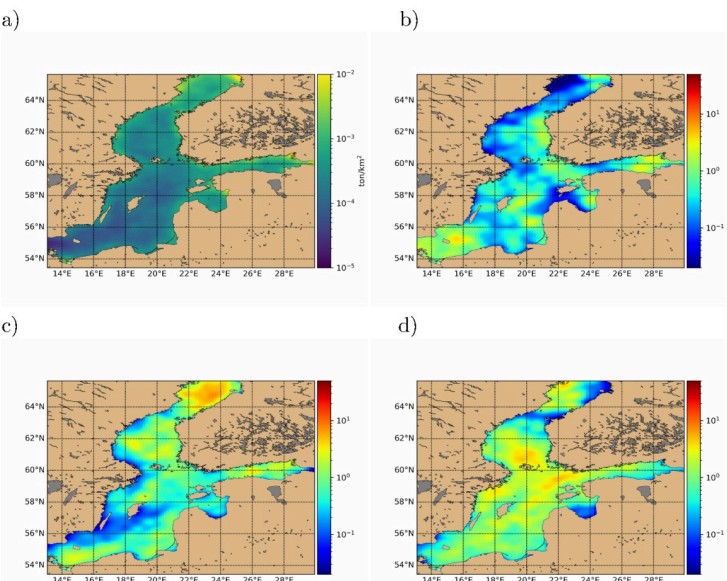

**Fig 6. Time-averaged macroplastic distributions in the Baltic Sea.** (a) Full average concentration (tons/km$^2$) for the 4 year period 2009–2012. Sub figures b, c, and d are seasonal averages, plotted relative to long term average distribution shown in a). The ratio is evaluated pixel by pixel, i.e. the maps are the local seasonal average divided by the local long term average. (b) Spring quarter 2012. (c) Summer quarter 2012. (d) Summer quarter 2011.

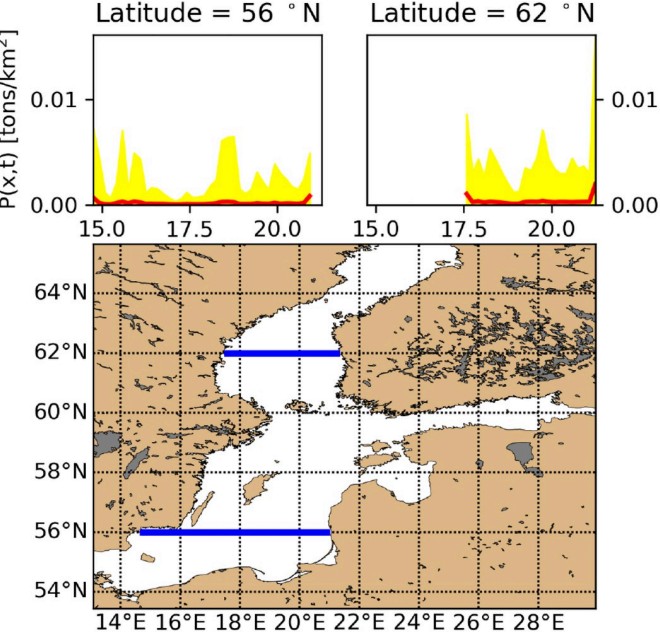

**Fig 7. Time statistics of plastic concentration $P(x, t)$ along the West-East transects at 56˚N and 62˚N, respectively, indicated in the lower figure by blue lines.** The upper left and upper right figure show time statistics for the period 2009–2012. The independent axes in the upper left and right figure correspond to the blue transects in the lower figure. Red full curves are average plastic concentration for baseline run, yellow area corresponds to +/- one standard deviation, at each location.

that the time variation of litter concentration is generally much larger than the average. In these two transects, the average time standard deviation is up to 7 times the average.

## Green's functions for pollution sources

In Fig 8 we show the Green's functions for river sources Luga (discharging approximately 4% of the Baltic riverine input) and Wisla (discharging approximately 2% of the Baltic riverine input). The distributions were generated as 4 year averages 2009–2012 with 2008 being spin-up year, using the DRRS equilibration scheme. We see that these two sources have very different areas of influence. The Luga pollution affects mostly the Bothnian Sea, whereas Wisla pollution affects most of the Eastern Baltic Sea. Clearly from a pollution management perspective it is of great importance to develop an understanding of the range of the pollution source Green's functions as we have done here. The simplest advection-diffusion-dissipation reference systems are those in one or two dimensions assuming constant parameters for advection ($v$), diffusion ($D_h$) and dissipation rate ($\lambda$), as summarized in supplementary material. Here,

a)

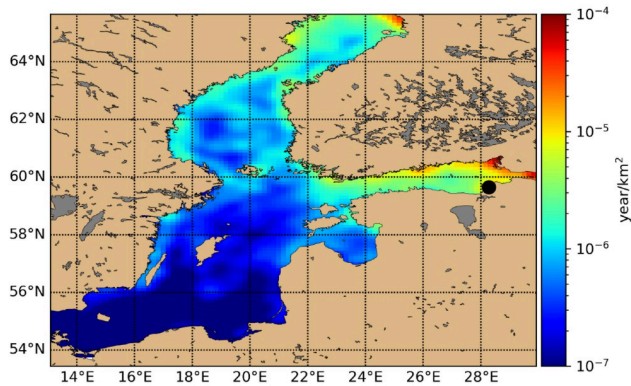

b)

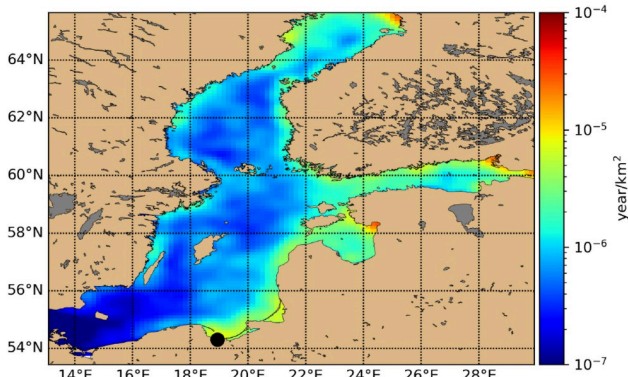

**Fig 8. Macro plastic Green's functions (in implied unit year/km², by Eq 11) for (a) river Luga (river mouth indicated by black dot) and (b) river Wisla (river mouth indicated by black dot).** All distributions are generated as 4 year averages 2009–2012 with 2008 being spin-up year, using the DRRS equilibration scheme.

the Green's function exhibits an exponential tail with decay length $\sqrt{D_h/\lambda}$ in the diffusive limit or $\sqrt{v/\lambda}$ in the advective limit, or an intermediate expression, so that the dynamic regime is characterized by the dimensionless number $\chi = \frac{v^2}{D_h\lambda}$ Interestingly, inserting typical values for current velocity $v \sim 0.3$ m/s and $D_h \sim 30$ m$^2$/s for the Baltic Sea (or other seas), suggests that $\chi$ is of order unity thus being on the boundary of dynamical regimes, and indicating that a variety of Green's function tail range scalings could be observed. In Fig 9 we show the corresponding Green's functions plotted against the (direct) distance to the source, again for rivers Luga and Wisla. The bivariate distributions have been smoothed using a non-parametric kernel density estimate. For river Luga in Fig 9(a) the expected exponential tail is clearly visible, corresponding to a range of 150 km. However for the river Wisla Fig 9(b), an exponential tail is not as apparent, thus this Green's function is not really canonically confined. For other sources in the Baltic, both these localized and extended types are seen. A more advanced analysis would also consider a water-bound non-direct distance.

## Transport mechanisms

The equation of motion Eq 3 has three advective components: water currents, Stokes drift and net direct wind drag, where the latter is expressed via the windage coefficient $k$ as $u_w = k\, u_{10}$. As argued, $k$ is different for each piece of macro plastic, partly reflecting $A_a \neq A_w$ in Eq 1, and in this work we used a representative value $k \sim 0.07$, which includes surface layer drag correction. It is of interest to explore in what range of $k$ values wind drag becomes the dominating advective component. We define the windage dominance index variability at a given time as

$$\delta_w(k; x, t) = \frac{k|u_{10}(x, t)|}{k|u_{10}(x, t)| + |u_c(x, t)| + |u_s(x, t)|} \quad (12)$$

and then the time average $\delta_w(k) = <\delta_w(k;x, t)>_{x,t}$. This is an Eulerian index, since it is not weighted by particle densities in the model domain. In Fig 10 we show the windage dominance index for the Baltic Sea averaged for the period 2008–2012. The colored area indicate the zone of normal variability in the physical forcing (average +/- standard deviation) of $\delta_w(k)$. We see that there is a transition to windage driven dynamics in the range $k \in [0.03; 0.05]$, and thus for the representative value $k \sim 0.07$ used in this work wind drag is often the dominating advective force. As the transition zone for $k$ is in the middle of the relevant range $k \in [0; 0.2]$, this means that highly variable transport rates are feasible for light and heavy plastic in practice, as advective vectors $u_c$, $u_s$, $u_w$ are not parallel, and their magnitude varies significantly, as the variance band in Fig 10 indicates.

## Discussion

Our results reveal new insights into the dynamics of floating plastic in the Baltic Sea and suggest highly variable dynamics and distribution of visible plastic in the Baltic Sea. As in all modelling studies parameter uncertainty limits the accuracy of the estimates, but predictions can be assumed to reflect the state of the art of our knowledge. In this exercise a number of sources of uncertainty apply some of which are unique to the modeling of micro and macro-plastic transport.

For example uncertainty in parameter weight-per-item (w) mainly changes the absolute normalization level of the spatial plastic distribution, whereas the qualitative trends are similar. The estimated abundances are quite sensitive to removal rate λ as well and therefore future attention should be devoted to refining the conversion (w) and removal parameterization. The removal rate λ is an ill-defined umbrella parameter representing complex and variable

a)

b)

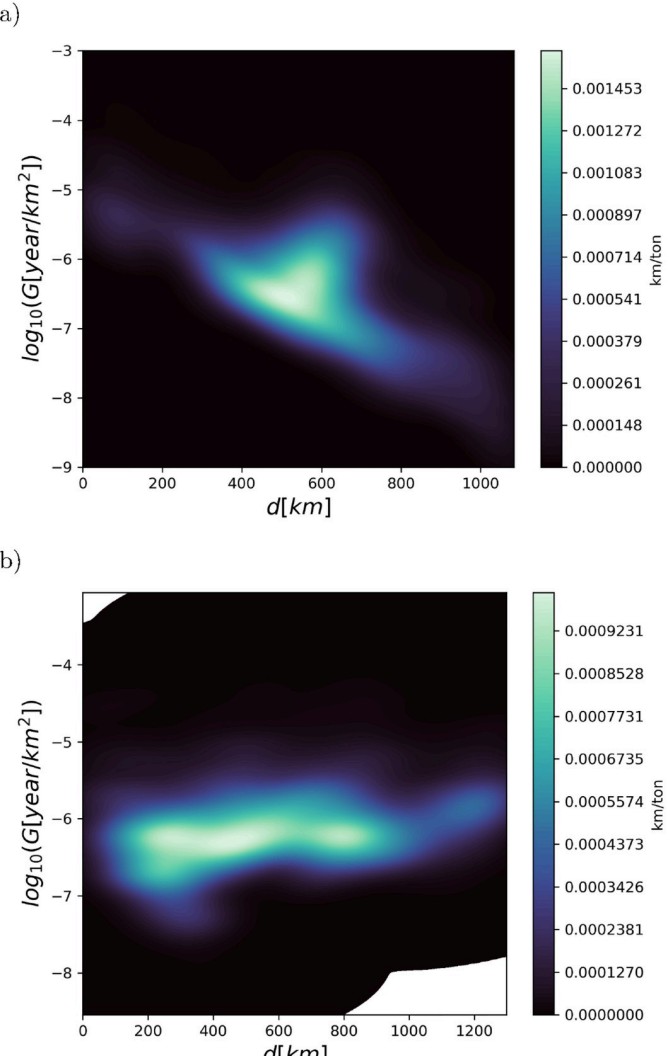

**Fig 9. Green's function density plotted against distance from the source for (a) river Luga (river mouth indicated by red dot) and (b) river Wisla (river mouth indicated by red dot).** All distributions are generated as 4 year averages 2009–2012 with 2008 being spin-up year, using the DRRS equilibration scheme. Bivariate distributions data is smoothed using a non-parametric kernel density estimate.

biofouling processes. It is believed to be a dynamic rate, depending significantly on local physical conditions and the abundance of macro and micro-algae and other potential settling organisms like bivalves, accumulated over the drift time of the plastic item. A clear seasonal trend is also anticipated in the biofouling rate. Further other sink processes need to be assessed to demonstrate the believed dominance of biofouling as determining eventual removal of marine litter.

Different macro litter fractions will likely display different dispersal patterns away from the source; this is mainly due to different experienced wind drag $k$, caused by buoyancy differences and differences in size and shape, but wave interaction and small scale surface circulation patterns (e.g. Langmuir circulations) may also play a significant role. It has already been pointed out that this has the potential to create spatial litter stratification and orientation by windage [10], which conversely would constitute a rich validation data set. Additionally the removal

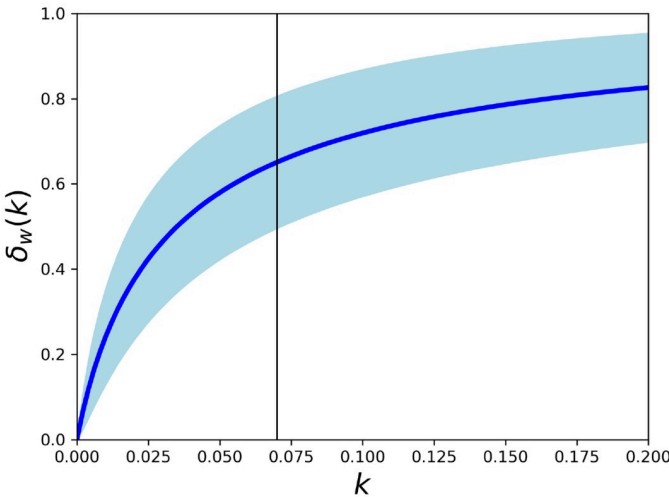

**Fig 10. Windage dominance index $\delta_w(k)$ (time and space average of Eq 12 as function of effective windage coefficient $k$ (Eq 1 for the Baltic Sea for the period 2008–2012.** The colored area indicate the statistical range of $\delta_w(k; x, t)$ over time and model domain, reflecting the natural variability range of $\delta_w(k; x, t)$. The vertical black line indicates the reference value $k = 0.07$ applied in the simulations.

rate $\lambda(x, t)$ determines the extent of the Green's function and will depend on mainly initial buoyancy and size of the objects (determining the level of biofouling needed to establish negative buoyancy leading to sinking), in addition seasonal and spatial differences in biofouling rate, which also depends of plastic type and surface texture (and possibly photo absorbance properties, determining effective temperature of biofouling substrate).

To confine uncertainty in transport simulations it is necessary to know the composition of litter from sources, expressed as a statistical distribution over $(k, \lambda)$ (as a first approximation). This is akin to the new trait-based paradigm in ecology, where the analysis focuses on key traits rather than individual species [52]. Assessing the statistical distribution over $(k, \lambda)$ of marine litter today is far beyond survey recording practices, where observational categories are very coarse, disjoint and aimed at low-effort postprocessing, e.g. "plastic", "plastic<5cm". As a minimum, to allow data-pooling and meta-studies, a common set of observational categories needs to be established, preferably a hierarchical system as applied for Habitat Classification [53, 54], allowing for maximum usages of all data with different degrees of detail recording. The idea of considering low-dimensional statistical distributions of litter types, rather than arbitrary more or less representative selections of specific litter pieces has already been suggested by [55], which suggested size, Corey shape factor and density as pragmatic covariates for the statistical distributions. Our work suggest that the most relevant covariates aligned with a transport modelling perspective are more precisely $(k, \lambda)$, which however are more complicated to measure routinely. Hence an important future research issue is linkage functions from easily measurable litter attributes (like size, shape indices and density) to the principal axes $(k, \lambda)$, and identifying further easily measurable litter attributes in addition to size, shape and density that determines the mapping to $(k, \lambda)$, like surface texture. Such input is important to standardize the sampling methodology and optimize reporting practices for detection, quantification, and characterization of plastic debris in the marine environment, which is critically lacking today [1]. By linearity of transport, the effective Green's function for a source is just $G_{k,\lambda}$ weighted by the statistical distribution over $(k, \lambda)$ of litter from that source.

The approach until now has not addressed that plastic litter possibly break up before eventual removal. The fragmentation time scale is believed to be decadal in typical situations [18]

and this is beyond the Green's function equilibration time scale. Breakup products will disperse independently. If fragments keep $(k, \lambda)$ corresponding to the parent, the result will be the same, because Lagrangian/Eulerian simulations have a diffusive term representing the effect of sub scale eddies statistically. According to [49] this is likely not the case for $\lambda$, and due to the air flow profile vertical scaling near the sea surface, it is likely not the case for $k$ neither. So studying the temporal dynamics of a distribution over $(k, \lambda)$ is of interest. also to address issues of long-term fate of marine litter at global scale.

The plastic source map applied in this work represents current best knowledge of land-based sources, however this data set has certain shortcomings, most importantly that the Russian sources are not available. We expect inclusion of this will lead to higher plastic concentrations in the Gulf of Finland and the Bothnian Sea. Land-based sources are believed to account for 70–80% of the marine litter [56, 57]. The remaining 20–30% of marine litter is believed to originate from nautical activity, most importantly lost and discarded fishing gear (ALDFG). It is estimated with high uncertainty that ALDFG constitutes 10–20% of the volume of total marine litter [58], however this fraction imposes a high mortality on fish. The HELCOM Baltic Marine Environment Protection Commission states that dolly ropes are not used in the bottom trawls used in the greater Baltic Sea [59]. We currently have no direct evidence on the spatial distribution of fishing gear loss, which would be required to be included in the source map Fig 1. A simple approach reflecting the uncertainty will be to add a homogeneous background concentration, representing lost fishing gear. In certain regional cases with open records on fishing activity it would be possible apply a source weighting based on fishing activity by location, time and fishing metier. In addition to fishing other nautical activities (e.g. shipping, ferrying and leisure), offshore platforms, aquaculture and abandoned vessels contribute to marine litter pollution, but the relative shares are currently unassessed. However, since this is a minor fraction of sources, we believe this will qualitatively conserve the contrasts that we have reported. A targeted study addressing this problem will be highly relevant to confirm this assumption. The Green's functions computed for riverine sources are insensitive to error and biases in the source map, but average plastic distribution is sensitive to source maps errors. It is important to stress that our setup will also work for an amended source map and results will be qualitatively similar, even though quantitative results may change a little reflecting updated input.

In this study we have applied constant litter influx, corresponding to the limited available data, without a potential seasonal pattern or short term interannual trends. However, it is not unlikely that a significant time variation is present in the influx for each source, reflecting e.g. precipitation dynamics, changing seasonal human consumption patterns and overall economic activity level etc. If such submodels were available, they could be applied as modulations on the constant litter influx levels applied in current setups. However, the interpretations will be tentative due to the assumption of stationarity in the derivation of the DRRS algorithms, and strictly that the full time dependent Green's function should be resolved rather than applying the DRRS algorithm for making the sampling more efficient. In Eqs 4–9 we developed the DRRS algorithm as a time-dependent scheme; Strictly the algorithm is only valid for stationary sources. This is seen by assuming that a particular source influx increased abruptly at a certain time; then the renormalization factor $R_T$ would reflect this increase correspondingly after the smoothing time scale $T$; this indicates that time scale $T$ should be chosen at or longer than the system equilibration time scale $(1/\lambda)$, and that the algorithm may be able to properly represent source variations with time scales longer than the system equilibration time. This is not really an issue in our case, since we just have stationary input sources, whereas the drift-export at the open Western entry to the Baltic (corresponding to last term of Eq 5) may fluctuate at a short time scale. We find in practice that $R_T(t)$ is quite stable, if $T$ is chosen longer than the

fluctuation time scale of the drift-export, and that the results $P(x, t)$ or $G(x, t)$ depend little on this. We also note that system interior transport events (i.e. garbage waves) are realistically represented, as they do not affect the input Eq 5 and thus not the normalization factor Eq 9, and therefore the variance bands in Fig 7 is argued to be representative.

Related to this discussion is also the effect of short-term fluctuations in litter distribution. A first generalization of the current framework in this direction is to simulate local temporal fluctuations $\sigma_P(x)$ in litter density with an equation similar to Eq 11, developed from the time dependent Green's function. This allows for assessing local ranges in plastic concentration and thus flagging whether e.g. maximum thresholds are exceeded temporarily, and the approximate duration and time fraction of threshold passing.

We have applied a 5 year time series of hydrography, which is beyond the equilibration time scale expected for Green's functions. The potential advantage of using real time series of transport fields over climatologies or cycled time series is an interesting issue not explored in this work. Using climatologies or cycled time series for drift-vector input is a common approach applied for centennial simulation timescales, because such long time series of drift-vector fields can not be established. This may impart the results in several ways; first, the output average of a nonlinear model does not correspond to the average input; secondly, real time series may give more realistic time series correlations in plastic abundance dynamics.

Our simulations were able to reproduce major spatial contrasts in average beach litter density for 2012–2016 recorded by the HELCOM monitoring program [51]. Ideally, simulations for the comparison should span same period. However, since no change in surface current, wave or wind statistics between periods 2009–2012 and 2012–2016 have been reported for the Baltic Sea [60] and further input sources are not time resolved, this would not lead to a significant change within the overall uncertainty of our study. If included, any minor overall increase in source influx from 2009–2012 to 2012–2016 would be absorbed in the beach affinity parameter $a$ fitted at the model/observation comparison, since litter transport processes are linear.

This study also identifies several issues, where data collection and environmental monitoring could be improved in the future to mature and enhance the realism of the framework. These are as follows. First, in the light of the spatio-temporal dynamics of the plastic addition to the system, seasonality of sources (beach and riverine) should be considered as currently only time-averaged sources are available. A first step would be to assess the approximate impact using an assumed seasonality to determine whether this should be addressed in future observational programmes. Secondly, representation of beach buffering effects should be improved, and research on this is currently being undertaken. Subsequently, compared to other materials being transported in the marine environment, visible plastic has several interesting issues in relation to aggregation potential which need to be examined. Since plastic (in many cases) has positive buoyancy most of the residence time in the marine environment, transport fields are not divergence free. Positive buoyancy combined with convergence is believed to be the main formation mechanism for blue-water plastic patches. Here, opposing wind, Stokes drift and surface current patterns (e.g. Langmuir circulations) may contribute to these local accumulation zones, in addition to boundary effects (beaches). Finally, residence time in the different domains is variable making the transport problem non-conservative and will contribute to spatial gradients in the output with significant research needed to understand and parametrize this critical issue.

## Conclusions

We have presented a new Lagrangian approach based on dynamical rescaling of weights-per-individual, the DRRS algorithm, to resolve the average plastic distribution, as well as its time

variation, in an open source-sink system, the Baltic Sea, where local sources have been mapped. The model setup has been validated satisfactorily against beach litter observations, with major spatial trends being reproduced. The model setup spans 2008–2012 as a representative multiannual time window, and includes realistic drift-vector fields from wind, waves and surface currents. The DRRS algorithm can equally well be applied to determine Green's functions (pollution plumes per source input) for individual sources, e.g. rivers. We found significant variation in the range of the tails of these Green's functions, and therefore explicit resolution of Green's functions is necessary rather than application of assumed pollution plumes based on a generic range estimate. We find a significant temporal variability in the plastic concentration locally, up to 7 times the average at considered transect lines, and this needs to be taken into account when assessing the ecological impact of marine litter, when sensitivity to litter is non linear, e.g. having a certain threshold. Since the dimensionless number $\chi = \frac{v^2}{D_h \lambda}$ is of order unity, we expect a range of Green's function tail scalings to be varying at regional scale, again stressing the importance of conducting actual litter transport simulations when planning litter cleanup mitigation investments. Finally we determined the ranges of windage $k > 0.1$ for floating plastic, where the dynamics are directly wind driven, as opposed to indirectly by surface currents and waves, and since the relevant range for $k$ is $[0; 0.2]$, we expect litter sorting by transport processes to be observable.

## Supporting information

**S1 File.**
(PDF)

## Author Contributions

**Conceptualization:** Asbjørn Christensen, Michael St. John.

**Data curation:** Jens Murawski, Jun She.

**Formal analysis:** Asbjørn Christensen.

**Funding acquisition:** Michael St. John.

**Investigation:** Asbjørn Christensen, Jens Murawski, Jun She.

**Methodology:** Asbjørn Christensen.

**Resources:** Michael St. John.

**Software:** Asbjørn Christensen.

**Validation:** Asbjørn Christensen, Jun She.

**Visualization:** Asbjørn Christensen.

**Writing – original draft:** Asbjørn Christensen, Jens Murawski, Jun She, Michael St. John.

**Writing – review & editing:** Asbjørn Christensen.

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
