## [Decision Letter · Decision Letter 0]

29 Sep 2022

PONE-D-22-21796Simulating transport and distribution of marine macro-plastic in the Baltic SeaPLOS ONE

Dear Dr. Christensen,

Thank you for submitting your manuscript to PLOS ONE. After careful consideration, we feel that it has merit but does not fully meet PLOS ONE’s publication criteria as it currently stands. Therefore, we invite you to submit a revised version of the manuscript that fully addresses all the points raised during the review process.

 Please submit your revised manuscript by Nov 13 2022 11:59PM. If you will need more time than this to complete your revisions, please reply to this message or contact the journal office at plosone@plos.org. Please include the following items when submitting your revised manuscript:A rebuttal letter that responds to each point raised by the academic editor and reviewer(s). You should upload this letter as a separate file labeled 'Response to Reviewers'.A marked-up copy of your manuscript that highlights changes made to the original version. You should upload this as a separate file labeled 'Revised Manuscript with Track Changes'.An unmarked version of your revised paper without tracked changes. You should upload this as a separate file labeled 'Manuscript'.

We look forward to receiving your revised manuscript.

Kind regards,

João Miguel Dias, Ph.D.

Academic Editor

PLOS ONE

Journal Requirements:

"This work was supported by the European Commission (https://ec.europa.eu) under the Horizon2020 programme as project CLAIM (Cleaning Litter by developing and Applying Innovative Methods in european seas), grant agreement No. 774586. The contribution of all authors (AC,JM,JS,MstJ) were funded by this grant."

3. We note that Figures 1, 4, 5, 6, 7 and 8 in your submission contain map/satellite images which may be copyrighted. All PLOS content is published under the Creative Commons Attribution License (CC BY 4.0), which means that the manuscript, images, and Supporting Information files will be freely available online, and any third party is permitted to access, download, copy, distribute, and use these materials in any way, even commercially, with proper attribution. For these reasons, we cannot publish previously copyrighted maps or satellite images created using proprietary data, such as Google software (Google Maps, Street View, and Earth). For more information, see our copyright guidelines: http://journals.plos.org/plosone/s/licenses-and-copyright.

a. You may seek permission from the original copyright holder of Figures  1, 4, 5, 6, 7 and 8 to publish the content specifically under the CC BY 4.0 license.  

Reviewers' comments:

Reviewer's Responses to Questions

**Comments to the Author**

1. Is the manuscript technically sound, and do the data support the conclusions?

Reviewer #1: Yes

Reviewer #2: Yes

2. Has the statistical analysis been performed appropriately and rigorously? 

Reviewer #1: I Don't Know

Reviewer #2: I Don't Know

3. Have the authors made all data underlying the findings in their manuscript fully available?

Reviewer #1: Yes

Reviewer #2: No

4. Is the manuscript presented in an intelligible fashion and written in standard English?

Reviewer #1: Yes

Reviewer #2: Yes

5. Review Comments to the Author

Reviewer #1: Being not a modeler myself, i strongly suggest revision of this manuscript from someone more expert than me in ocean modeling. However, the article is well-written and no major issues were found from my side. Just a biologist’s note at Line 469: crustaceans are only a minor component of biofouling communities, indeed most of the buoyancy changes are very likely due to intensive fouling by macro and micro-algae, not crustaceans. I suggest to remove or change this.

Reviewer #2: The present work described a new Lagrangian approach based on dynamical rescaling of weights-per-individual, the DRRS algorithm, to resolve the average plastic distribution, as well as its time variation, in an open source-sink system, the Baltic Sea, where local sources have been mapped. In general, the use of the model to simulate plastic in the water is interesting and useful to evaluate the potential influence of plastics.

Some specific comments:

I have some doubts about the methods. The hydrodynamic modelling plan is not clear. Parameters and detailed modelling processing used in this study are suggested. Would you describe the process of this simulation experiment in detail?

Add more specifications of the study area, and include a new subsection.

What is the database used for the identification of the macro litter sources?

Explain better the assumption of the 4g/item (line 147).

Line 167-168: What is the uw? This parameter is not mentioned in the text.

The section on the material methods is more confusing. I suggest authors try to be more concise.

The first paragraph of the results section should be on the methods (lines 351-362). Idem for lines 420-425. The results section must be restructured.

The authors only considered constant litter influx (line 538), may the authors should also discuss the contribution by other sources if considered. The authors should supply information on land use around the bay and discuss other pathways of plastic pollution.

Please insert the color bar percentage in Figure 1b.

In figure 6 the color bars have to be adjusted so that the same color scale is used in all panels.

6. PLOS authors have the option to publish the peer review history of their article (what does this mean?). If published, this will include your full peer review and any attached files.

Reviewer #1: No

Reviewer #2: No

---

## [Author Response · Author response to Decision Letter 0]

14 Dec 2022

>>> For a more nicely formatted response, please use the file Response_to_Reviewers.pdf <<<

response to reviewer 1 ===============================

no major issues were found from my side. Just a biologist’s note at Line 469: crustaceans are only a

minor component of biofouling communities, indeed most of the buoyancy changes are very likely due

to intensive fouling by macro and micro-algae, not crustaceans. I suggest to remove or change this.

---

The sentence:

It is believed to be a dynamic rate, depending significantly on local physical conditions, planktonic

dynamics, and abundance of crustacean settlers accumulated over the drift time of the plastic item.

was replaced with this:

It is believed to be a dynamic rate, depending significantly on local physical conditions

and the abundance of macro and micro-algae and other potential settling organisms like bivalves,

accumulated over the drift time of the plastic item.

(I have seen several pictures of marine macro-plastic with bivalves attached).

response to reviewer 2 ===============================

3. Have the authors made all data underlying the findings in their manuscript fully available?

Reviewer #2: No

---

Unfortunately, the reviewer is not specific about what this exactly concerns. In a general sense, the data

underlying our paper is constituted by code, input data and output data.

Code: in Supporting information (section "S4 Software availability") we have in detail

accounted for the full and unrestricted access to the code underlying our reported findings. Our

code is distributed as open source at https://github.com/IBMlib/IBMlib .

Input data: physical forcing data applied in our study has been described fully in section

"Materials and methods"; we do not have distribution rights to the physical forcing data applied

and can consequently not supply this, but in "Materials and methods" it is made clear what

product is used and who the producer is, so they can be contacted by independent investigators.

Some of the forcing data (wind fields and Stokes drift) is open access, whereas some restricted

access (current fields). The plastic source mapping is available and that has now been

emphasized in section "S4 Software availability". All scripts for for baseline runs (Greens

functions and regional plastic distribution dynamics) are also now made available, and these

scripts generate input files for running simulations with parameter setting as described in

"Materials and methods" and Table 1. A README file is provided in a DDRS software folder

for the configuration applied in the current paper. All this has now been emphasized in section

"S4 Software availability" in a new paragraph "S4.3 Configuration for the Baltic case study".

Output data: these high resolution binary gridded data sets containing two space and one time

dimension are very large and unsuitable for redistribution. It is not customary that such

extensive binary data sets are published as part of a scientific publication. Since our code and

input data are fully described or provided, as outlined above, any independent investigator may

recreate the output data, if needed.

Consequently, we argue that our reported findings are strictly reproducible by anybody that wants to

reproduce our findings or extend our work.

I have some doubts about the methods. The hydrodynamic modelling plan is not clear. Parameters and

detailed modelling processing used in this study are suggested. Would you describe the process of this

simulation experiment in detail? Add more specifications of the study area, and include a new

subsection.

---

A new subsection has been added to "Materials and methods" that explicitly describes the baseline

simulations (Greens functions and regional plastic distribution dynamics) that underlies our paper. The

technical aspects of the simulations has already been described, as the reviewer acknowledges.

Key facts about the physical oceanography of the study area has already been provided in the first lines

of "Materials and methods"; we have extended this with a few details about biological oceanography

and geography, and distilled this out into a new subsection "Study area: the Baltic Sea" that should give

the reader a more comprehensive impression of the study area.

What is the database used for the identification of the macro litter sources?

---

We have now inserted the exact reference for the macro litter source data set applied in this study;

further we note that the macro litter source data set formatted to Lagrangian simulations is made

available to the scientific community, as described in the new paragraph "S4.3 Configuration for the

Baltic case study".

Explain better the assumption of the 4g/item (line 147).

---

This is not an assumption, but an observation that if w = 4 g/item then total riverine and coastline input

are equal in this data set. We add in this data set to emphasize that this threshold value applies to that

data set, and different threshold value must be expected for other data sets, and start the sentence with

We determine that ... to make statement clearer. As stated in the next sentence we apply w = 10 g/item

in our simulations, as supported by a reference, i.e. not w = 4 g/item (we added in our simulations to

make this clearer).

Line 167-168: What is the uw? This parameter is not mentioned in the text.

---

u_w designate drift velocity; this is now indicated in line 165, before first occurrence iin Eq. 1

The section on the material methods is more confusing. I suggest authors try to be more concise.

---

Some confusion emerge from an inconsistent usage of headings "Materials and methods"; this is now

amended. Now our "Materials and methods" consistent of these 8 sub sections:

\\subsection*{Study area: the Baltic Sea}

\\subsection*{Baltic Sea physical model}

\\subsection*{Baltic Sea macro litter sources}

\\subsection*{Baltic Sea macro litter transport model}

\\subsection*{The DRRS scheme for quasi-equilibrium distributions}

\\subsection*{Green's functions and plastic distribution}

\\subsection*{Baseline simulations}

\\subsection*{Model validation}

the latter two of which was suggested by the present reviewer. These sub sections represent a logical

progression and the modularity of our approach, and should aid the reader to zoom in on aspects of

interest. We hope the latest revisions meet to request of the reviewer, as this reviewer comment

unfortunately is not very specific. We also draw attention to the fact that our paper present

methodological progresses (a publication type warranted by the editorial guidelines), and therefore our

section is slightly longer than papers reporting results obtained by standardized approaches.

The first paragraph of the results section should be on the methods (lines 351-362). Idem for lines 420-

425. The results section must be restructured.

---

lines 351-362: we have followed the reviewers suggestion and moved the model validation paragraph

to the "Materials and methods" section.lines 420-425: Presentation style in natural sciences across journals and authors is quite diverse; the

only apparent consensus in writing guides is that the premises of the results should be fully reported in

"Materials and methods" (with possible links to supp. material), so that presented results are completely

reproducible by independent investigators. The content of lines 420-429 is not methods used to

produce material to the section "Results", but reflections on the patterns of material in the section

"Results"; therefore we feel that this content belongs here. We have inlined the previous equation 12, so

that it is more clear that it is part of our observations on the simulation output rather than part of the

method description. We browsed the recent literature in different journals and quickly found 3

examples of papers with reflection equations is sections Results/Discussion:

Ecological Modelling 474 (2022) 110153 (page 4)

https://doi.org/10.1098/rstb.2020.0414

https://doi.org/10.1371/journal.pone.0278167 (result dsicussion)

However, if the editor agree with the reviewers comment on lines 420-425, we are willing to move this

to the sub section "Green's functions and plastic distribution", even though we think it is slightly

misplaced there and will make the "Materials and methods" section bulkier.

Further, to enhance the overview of our presented results, we have subdivided the "Results" section

into 3 sub sections,

\\subsection*{Macroplastic dynamics in the Baltic Sea}

\\subsection*{Green's functions for pollution sources}

\\subsection*{Transport mechanisms}

The authors only considered constant litter influx (line 538), may the authors should also discuss the

contribution by other sources if considered. The authors should supply information on land use around

the bay and discuss other pathways of plastic pollution.

---

The applied plastic sources maps are supposed to represent all land-based sources

In lines 519-537 we extensively discussed abandoned, lost and discarded fishing gear (ALDFG),

another important recognized source of plastic pollution not included in the source map applied. We

have extended this section with other sources including other nautical activity (e.g. shipping, ferrying

and leisure), offshore platforms, aquaculture and abandoned vessels, added references for the ratio of

land/sea-based pollution and suggested future amendments. Even though land use is an interesting

subject, we find a digression on land use around the bay (Baltic Sea?) to be beyond the scope of our

paper as only the net input is of importance for our study (not why) and we prefer readers interested in

this topic consult the relevant literature.

Please insert the color bar percentage in Figure 1b.

---

Figure 1b is only a schematic view, reproduced from another source. The arrows represents large-scale

water mass transport, and a scale-bar for volume transport is not available from the source. We have

now noted in the figure caption that arrows represent a schematic view.

In figure 6 the color bars have to be adjusted so that the same color scale is used in all panels.

---

Fig. 6a compared to Fig. 6b,c,d show different things, and therefore it is not appropriate to apply same

color scale. Fig 6a is absolute concentration, whereas Fig. 6b,c,d are concentration relative to the local

long-term average concentration. We find this way of plotting more informative, as it better display

contrasts. Value range of Fig. 6a compared to Fig. 6b,c,d is very different, and therefore applying same

scale (even for a logplot) will remove much of the contrast in all figures. We have reformulated the

caption so the difference between Fig. 6a and Fig. 6b,c,

---

## [Decision Letter · Decision Letter 1]

5 Jan 2023

Simulating transport and distribution of marine macro-plastic in the Baltic Sea

PONE-D-22-21796R1

Dear Dr. Christensen,

We’re pleased to inform you that your manuscript has been judged scientifically suitable for publication and will be formally accepted for publication once it meets all outstanding technical requirements.

Kind regards,

João Miguel Dias, Ph.D.

Academic Editor

PLOS ONE

Additional Editor Comments (optional):

Reviewers' comments:

Reviewer's Responses to Questions

**Comments to the Author**

1. If the authors have adequately addressed your comments raised in a previous round of review and you feel that this manuscript is now acceptable for publication, you may indicate that here to bypass the “Comments to the Author” section, enter your conflict of interest statement in the “Confidential to Editor” section, and submit your "Accept" recommendation.

Reviewer #2: All comments have been addressed

2. Is the manuscript technically sound, and do the data support the conclusions?

Reviewer #2: Yes

3. Has the statistical analysis been performed appropriately and rigorously? 

Reviewer #2: Yes

4. Have the authors made all data underlying the findings in their manuscript fully available?

Reviewer #2: (No Response)

5. Is the manuscript presented in an intelligible fashion and written in standard English?

Reviewer #2: Yes

6. Review Comments to the Author

Reviewer #2: (No Response)

7. PLOS authors have the option to publish the peer review history of their article (what does this mean?). If published, this will include your full peer review and any attached files.

Reviewer #2: No

---

## [Editor Report · Acceptance letter]

6 Jan 2023

PONE-D-22-21796R1 

Simulating transport and distribution of marine macro-plastic in the Baltic Sea 

Dear Dr. Christensen:

I'm pleased to inform you that your manuscript has been deemed suitable for publication in PLOS ONE. Congratulations! Your manuscript is now with our production department. 

Kind regards, 

on behalf of

Prof. João Miguel Dias 

Academic Editor

PLOS ONE